



# Differing precipitation response between Solar Radiation Management and Carbon Dioxide Removal due to fast and slow components

Anton Laakso[1,2], Peter K. Snyder[1], Stefan Liess[3], Antti-Ilari Partanen[4] and Dylan B. Millet[1]

[1]Department of Soil, Water and Climate, University of Minnesota, Twin Cities, St. Paul, MN-55108, Minnesota, USA
[2]Finnish Meteorological Institute, Atmospheric Research Centre of Eastern Finland, Kuopio, FI-70200, Finland
[3]Department of Earth Sciences, University of Minnesota, Minneapolis, MN-55455
[4]Finnish Meteorological Institute, Climate System Research, Helsinki, FI-00100, Finland

*Correspondence to*: Anton Laakso (anton.laakso@fmi.fi)

**Abstract.** Solar Radiation Management (SRM) and Carbon Dioxide Removal (CDR) are geoengineering methods that have been proposed to prevent climate warming in the event of insufficient greenhouse gas emission reductions. Here, we have studied temperature and precipitation responses to CDR and SRM with the RCP4.5 scenario using the MPI-ESM and CESM Earth System Models (ESMs). The SRM scenarios were designed to meet one of the two different climate targets: to keep either global mean 1) surface temperature or 2) precipitation at the 2010-2020 level via stratospheric sulfur injections. Stratospheric sulfur fields were simulated beforehand with an aerosol-climate model, with the same aerosol radiative properties used in both ESMs. In the CDR scenario, atmospheric $CO_2$ concentrations were reduced to keep the global mean temperature at approximately the 2010-2020 level. Results show that applying SRM to offset 21st century climate warming in the RCP4.5 scenario leads to a 1.42% (MPI-ESM) or 0.73% (CESM) reduction in global mean precipitation, whereas CDR increases global precipitation by 0.5% in both ESMs for 2080-2100 relative to 2010-2020. In all cases, the simulated global mean precipitation change can be represented as the sum of a slow temperature-dependent component and a fast temperature-independent component, which are quantified by regression method. Based on this component analysis, the fast temperature-independent component of $CO_2$ explains the global mean precipitation change in both SRM and CDR scenarios. Based on the SRM simulations, a total of 163-199 Tg(S) (CESM) or 292-318 Tg(S) (MPI-ESM) of injected sulfur from 2020 to 2100 was required to offset global mean warming based on the RCP4.5 scenario. To prevent a global mean precipitation increase, only 95-114 Tg(S) was needed and this was also enough to prevent global mean climate warming from exceeding 2 degrees above preindustrial temperatures. The distinct effects of SRM in the two ESM simulations mainly reflected differing shortwave absorption responses to water vapor. Results also showed relatively large differences in the individual (fast versus slow) precipitation components between ESMs.

## 1 Introduction

It is now widely recognized that fast greenhouse gas (GHG) emission reductions, especially for carbon dioxide ($CO_2$), are needed if ongoing global warming is to be minimized. The Intergovernmental Panel on Climate Change (IPCC) Special Report



on Global warming of 1.5 °C (henceforth SR15, IPCC, 2018) brought to wider attention many, partly irreversible, risks associated with global mean warming of 1.5 °C above the pre-industrial level (Liu et al., 2018; Schleussner et al., 2016; Seneviratne 2018). The aim of the 2015 Paris Agreement was to maintain the global mean temperature within 2 °C of the pre-

industrial level, and to pursue efforts to limit the mean increase to 1.5 °C (UNFCCC, 2015). Based on climate model simulations, only one of the Representative Concentration Pathway (RCP) scenarios used in the IPCC Fifth Assessment report (IPCC, 2014) is associated with global mean warming of less than 2 °C; this pathway includes presumptive mitigation after year 2010, which has not taken place (van Vuuren et al., 2007). Millar et al. (2017) and Rogelj et al. (2018) have shown that limiting warming to 1.5 °C is still possible, but would require rapid abandonment of fossil fuel and a reduction of energy use.

In addition, air quality legislation will likely lead to decreased cooling from anthropogenic aerosols, which might by itself be enough to increase global mean temperatures over the 1.5 °C target (Hienola et al., 2018).

Geoengineering methods have been proposed to prevent dangerous climate warming if $CO_2$ emissions are not reduced quickly enough (e.g., Caldeira et al., 2013). Such techniques are usually divided into two categories. One is Carbon Dioxide Removal

(CDR), whereby $CO_2$ is removed from the atmosphere—thus addressing the root cause of climate warming (Royal Society, 2009). While these actions will face many political, economic and technical challenges they are most likely needed in some form to avoid 1.5 C warming (Luderer et al., 2018, IPCC 2018. The second is Solar Radiation Management (SRM), which aims to increase the shortwave (solar) reflectivity of the atmosphere or Earth's surface. The Paris agreement states that the 2 °C target should be achieved by reaching a balance between anthropogenic GHG emissions and anthropogenic GHG sinks

(i.e., CDR) (UNFCCC, 2015). However, challenges related to mitigation and CDR, underestimation of future carbon budgets, or new scientific understanding of tipping points could lead to increased interest in using SRM to avoid crossing the Paris Agreement temperature thresholds.

Discussions related to SR15 and the Paris Agreement have concentrated mainly on global mean temperature change, rather

than on regional variations in temperature changes (Collins et al., 2018; Seneviratne et al., 2018) or on other climate impacts, such as changes in precipitation, that are driven by temperature changes or caused directly by GHG or other forcing agents. On the global scale, precipitation changes can be separated into a surface-temperature dependent slow component, which does not depend on the forcing agent causing the underlying temperature change, and a temperature-independent fast component, which is caused directly by the altered atmospheric radiation absorption (Bala et al., 2009; Myhre et al., 2017; Samset et al.,

2016). Changes to the hydrological cycle thus depend not only on the degree of warming but also on the forcing agents and emission changes that are causing the warming. As a result, different emission pathways can lead to different precipitation changes even if they result in similar global mean temperatures. Such hydrologic changes may have a larger impact on human wellbeing than changes in temperature due to impacts on floods, droughts, water resources and ecosystems (Lausier and Jain 2018).




Problems and side effects associated with SRM have been discussed extensively (Robock et al. 2009; Royal Society 2009). One fundamental problem is that SRM would decrease global mean precipitation through the direct radiative effect described above. This can be understood as follows. Given a GHG concentration increase, less outgoing longwave (LW) radiation escapes to space, causing surface temperatures to increase until a new equilibrium is achieved. SRM methods aim to offset

this temperature increase by reducing incoming shortwave (SW) solar radiation. Thus, even though the total radiative flux may be the same between an increased GHG + SRM scenario and the unperturbed climate, the atmospheric SW and LW radiative fluxes differ. This has been shown in general to lead to a decrease in global mean precipitation (Bala et al., 2008). In general, the suite of climate responses arising from a LW radiation change cannot be fully compensated by modifying SW radiation. Use of SRM thus involves a tradeoff between temperature and precipitation on the global scale.


CDR methods are considered less risky than SRM as these methods remove $CO_2$ from the atmosphere and thus reduce the atmospheric GHG concentration (Royal Society 2009). However, climate change is not a reversible process due to factors such as sea and glacier ice melt, sea level rise, and carbon cycle changes (Frölicher and Joos 2010; Wu et al., 2015). In addition, climate does not adapt immediately to a change in radiative forcing. For example, due to ocean thermal inertia global

temperatures will continue to change decades or even centuries after a given radiative forcing perturbation. It is therefore important that CDR scenarios be studied to assess climate responses beyond changes to global mean temperature.

In this study, the temperature and precipitation responses to CDR and SRM are simulated with two Earth System Models (ESMs). The mechanisms driving global mean precipitation changes are assessed by separately examining the temperature-

dependent slow response and radiatively-induced fast response for differing magnitudes of SRM and CDR. This methodology can be used to better understand impacts of CDR and SRM, and to predict total temperature and precipitation responses if CDR and SRM are used simultaneously. Unlike in several previous studies, here, fast and slow responses are quantified by a regression method instead of a fixed sea surface temperature (SST) method (Duan et al., 2018; Myhre et al., 2017; Samset et al., 2016). Advantage of regression method is that it separate totally temperature-dependent and independent responses while

in the fixed-SST method land temperature adjustments are included in the temperature-independent fast response. We also study regional disparities in temperature and precipitation responses for both geoengineering techniques, and estimate the $SO_2$ emission amounts required to keep either temperature or precipitation at present-day levels.

We simulate three geoengineering scenarios against a Representative Concentration Pathway 4.5 (RCP4.5) scenario (Thomson

et al., 2011). First, we examine two SRM scenarios designed to address two different climate targets: keeping either global mean 1) surface temperature or 2) precipitation at 2010-2020 level via stratospheric sulfur injections. Next, we examine a CDR scenario designed to keep the global mean temperature at approximately the 2010 level. We used an aerosol-climate model to simulate stratospheric aerosol fields, and two separate ESMs (MPI-ESM and CESM) to simulate the climate response to SRM and CDR.



## 2 Methods

### 2.1 Models

This study was conducted using three climate models: one aerosol-climate model with fixed sea surface temperature, and two ESMs. We first simulated stratospheric aerosol fields with the aerosol climate model ECHAM-HAMMOZ. We then implemented the radiative properties of these fields in two ESMs—the Max Planck Institute Earth System Model (MPI-ESM) and the National Center for Atmospheric Research's Community Earth System Model (CESM)—for simulation of the various scenarios. For each scenario, we run a three-member ensemble with both ESMs.

### 2.1.1 ECHAM – HAMMOZ

We defined the radiative properties of the aerosol fields resulting from stratospheric injections of sulfur dioxide ($SO_2$) with the MAECHAM6.1-HAM2.2-SALSA global aerosol-climate model (Bergman et al., 2012; Kokkola et al., 2008; Laakso et al., 2016; Stevens et al., 2013, Zhang et al., 2012). In this model, the ECHAM atmospheric module (Stevens et al., 2013) is coupled interactively with the HAM aerosol module (Zhang et al., 2012). The HAM module calculates the emissions, removal, and radiative properties of aerosols along with the associated gas and liquid phase chemistry. The model includes the SALSA explicit sectional aerosol scheme (Kokkola et al., 2018), which describes aerosols based on number and volume size distributions with 10 and 7 size sections for soluble and insoluble particles, respectively. The model simulates the microphysical processes of nucleation, condensation, coagulation, and hydration. The model was configured as described by Laakso et al. (2017). Simulations were performed at T63L47 resolution, which corresponds approximately to a 1.9° x 1.9° horizontal grid with 47 vertical levels reaching up to ~80 km. The model accurately simulates stratospheric aerosol loads and radiative properties based on observations of the 1991 Mt. Pinatubo eruption (Laakso et al., 2016; Kokkola et al., 2018). It should be noted that this model configuration does not simulate the quasi biennial oscillation at L47 resolution. Hydroxyl radical (OH), which impacts the oxidation of $SO_2$ to sulfate as well as ozone concentration, is accounted for through prescribed monthly mean fields.

### 2.1.2 MPI-ESM and CESM

MPI-ESM (Giorgetta et al., 2013) consists of the same atmospheric model (ECHAM6.1) as ECHAM-HAMMOZ, and the MPI-ESM simulations here also employed the same T63L47 resolution as the ECHAM-HAMMOZ simulations described above. MPI-ESM includes the JSBACH active land model (Reich et al., 2013) and the Max Planck Institute Ocean Model (MPIOM) (Junglaus et al., 2012), both fully coupled to the atmospheric module. MPIOM includes the HAMOCC ocean biogeochemistry model (Ilyina et al., 2013). The tropospheric aerosol climatology of Kinne et al. (2013) is used in all scenarios.



CESM version 1.2.2 (Hurrell et al., 2013) consists of the Community Atmospheric Model (CAM4) which is used with
horizontal resolution of 0.9° x 1.25° and 26 vertical levels up to 40 km (finite volume grid). It is coupled to the Parallel Ocean
Program (POP2) ocean model, the Community Land Model (CLM4) and Community Ice CodE (CICE4) sea ice model.

### 2.1.3 Implementing prescribed aerosol fields in ESMs

To examine the effects of solar radiation management by stratospheric sulfur injections we implemented prescribed sulfate
fields in the ESMs as described by Laakso et al. (2017). First, we used ECHAM-HAMMOZ to the simulate aerosol fields
resulting from gaseous $SO_2$ injections. These simulations include a 2-year spin-up period followed by a 5-year steady-state
period. From this 5-year period, mean stratospheric values for aerosol optical depth (AOD), single scattering albedo (SSA),
and the asymmetry factor (ASYM) were archived as monthly output in 14 SW bands plus 16 LW bands for (absorption) AOD.
We then implemented these fields in the two ESMs as prescribed zonal and monthly mean fields. ECHAM-HAMMOZ and
MPI-ESM share the ECHAM atmosphere model, which itself uses the Rapid Radiative Transfer Model. Because the same
resolution (T63L47) was employed for both ECHAM-HAMMOZ and MPI-ESM, the only differences in aerosol radiative
properties between the models was caused by zonal and monthly averaging of the radiative properties (however, the total AOD
did not vary between the two). In the case of CESM, aerosol fields from the ECHAM-HAMMOZ simulations had to be
interpolated horizontally to 0.9° x 1.25° and to 26 vertical levels. Because CAM4 uses different wavelength bands than does
ECHAM (7 LW bands and 19 SW bands), we interpolated the aerosol optical properties accordingly.


The above implementation ensures that SRM radiative effects are consistent in both ESMs, while also enabling longer-term
analyses since computationally expensive aerosol microphysics are prescribed rather than simulated online. The aerosol
radiative effects are nevertheless based on explicit simulations of aerosol microphysics and of the resulting aerosol size
distribution and spatial-temporal variability. Our methodology is therefore more physically realistic compared to approaches
that simply reduce the solar constant or apply idealized zonally homogenous aerosol fields. Realistic simulation of aerosol
microphysics is necessary for robust prediction of the associated radiative effects, which depend on the size, properties, and
location of the particle. In the stratosphere, particle life times are roughly one year, so that  microphysical processes such as
coagulation and condensation play a greater role than in the troposphere. As a result of these microphysical processes, radiative
forcing from stratospheric sulfur injections does not increase linearly with the amount of injected sulfur (Niemeier et al., 2015).

### 155  2.2 Simulations

To simulate SRM stratospheric aerosol fields, we performed six SRM and one control simulation with ECHAM-HAMMOZ.
Here, $SO_2$ was injected continuously throughout the simulation at 20 km altitude between 10 N and 10 S latitudes. Each of the
ECHAM-HAMMOZ simulations included injection of 1, 2, 3, 4, 5 or 6 Tg(S)/yr.





We divide the ESM simulations into two groups: 1) component analysis simulations, and 2) scenarios. Component analysis simulations are performed to enable subsequent separation of the slow (temperature-dependent) and fast (temperature-independent) responses to the specific forcing agent (Gregory et al., 2004) based on a regression method. In this method, an individual forcing agent ($CO_2$ or SRM) is added to the steady state climate conditions, and different climate variables are regressed against the global mean surface temperature change. The fast and slow responses for a specific forcing agent is then

obtained from the fitted regression line. Specifically, the fast temperature-independent response is derived as the intercept (zero temperature change), while the slow temperature-dependent response is derived as the slope. This analysis is done for three purposes: 1) to evaluate the implementation of the stratospheric aerosol fields across the two ESMs, 2) to quantify differences in radiative forcing and climate sensitivity between models under a specific forcing agent, and 3) to separate the fast and slow precipitation responses of the forcing agents. A total of 9 scenarios are simulated with both ESMs: preindustrial,

6 SRM scenarios with 1, 2, 3, 4, 5, 6 Tg(S) injections, and $2xCO_2$ and $4xCO_2$ conditions.

All component analysis simulations start from a radiatively balanced climate for pre-industrial conditions. A forcing agent ($CO_2$ or SRM) is introduced at the outset of the simulation while other conditions are kept pre-industrial levels. We simulated three 20-year ensemble members for each component analysis scenario in Table 1.


Scenario simulations were based on RCP 4.5 (Moss et al., 2010; van Vuuren et al., 2011), and included: i) one baseline scenario with no geoengineering (RCP45), ii) two SRM scenarios designed to keep global mean surface temperature (SRM-TEMP) or precipitation (SRM-PRECI) at 2010-2020 mean values, and iii) one CDR scenario designed to keep global mean surface temperature at the 2010-2020 mean value (CDR). In each case, three ensemble members were simulated for years 2010-2100.


In the RCP 4.5 scenario, radiative forcing stabilizes several decades before the end of the simulations (year 2100), leading to warming clearly below that seen in business as usual scenarios, but above the targets defined in the Paris Agreement. For the SRM-TEMP and SRM-PRECI scenarios, the global mean temperature or precipitation was kept close to the 2010-2020 mean via controlled stratospheric sulfur injections.


In practice, the SRM scenario objectives were achieved by adjusting the aerosol loading as needed based on the continuous $SO_2$ injection simulations from ECHAM-HAMMOZ (Sect. 2.1.3). Specifically, the SRM was controlled annually based on mean temperature or precipitation values from the two preceding years, as follows:

If ( ($X_{year-1}$ + $X_{year-2}$)/2 > $X_{2010-2020}$ + A ) then $SRM_{year}$ = $SRM_{year-1}$ + 1 Tg(S)/yr.
       If ( ($X_{year}$-1 + $X_{year}$-2)/2 < $X_{2010-2020}$ - A ) then $SRM_{year}$ = $SRM_{year-1}$ - 1 Tg(S)/yr,





where $X_{year-1} + X_{year-2}$ are the global mean temperature (for SRM-TEMP) or precipitation (for SRM-PRECI) in the two preceding years, while $X_{2010-2020}$ is the corresponding global mean value for 2010-2020 based on RCP45. An approximation

inherent in this approach is that transitory ramp-up and ramp-down periods in the stratospheric aerosol burden with 1 Tg(S)/yr changes in SRM are not taken account.

The A parameter is a boundary value and set to 0.2 K in SRM-TEMP, which based on our test simulations is generally larger than natural variability. For SRM-PRECI A is defined to correspond to a 0.5% change in the global mean precipitation in the

model. If both of the above conditions are false, the stratospheric sulfur injections are maintained at the prior year's level. SRM simulations are initialized with 1 Tg(S)/yr injections at year 2020. A two-year running window is used to avoid undue influence from natural variability in global mean temperature or precipitation. Use of a longer window is suboptimal because the temperature or precipitation change the year following an SRM adjustment then does not carry sufficient weight for the subsequent evaluation. This can lead to overly large temperature or precipitation changes before the need to act is recognized.


In the CDR scenarios, $CO_2$ removal was likewise initialized at year 2020. Here, the annual $CO_2$ increase based on RCP4.5 was counteracted by a 1% annual removal of the atmospheric $CO_2$ concentration. This process was continued until year-2070, when radiative forcing is stabilized in the RCP4.5 scenario. Accounting for both RCP4.5 emissions and CDR, the total atmospheric $CO_2$ concentration is then reduced yearly by 0.3 - 0.6 % between 2020-2070 (Fig. 1). Removing 1% of atmospheric $CO_2$ in

2020 corresponds to negative emissions of approximately 8.7 GtC/yr. As carbon cycle feedbacks (i.e., outgassing from natural carbon sinks) lower the efficiency of CDR (Tokarska and Zickfeld, 2015), the actual amount of sequestered carbon would in reality need to be even higher than this. Achieving such high negative emissions in 2020 would be virtually impossible. The rate required is higher than the maximum estimated sustainable potential of the highest-potential negative emission technologies (Fuss et al., 2018), without even considering competition between the methods. Among SR15 scenarios pursuing

the most aggressive CDR, the median carbon sequestration rate for the primary employed method (bioenergy with carbon capture and storage) reaches ~4 GtC/yr in 2100 (Rogelj et al., 2018). Thus, the CDR scenario employed here should be considered an idealized high-end carbon removal scenario, and we do not speculate how CDR could be achieved and do not study impacts of any specific CDR technology. All non-$CO_2$ GHG concentrations and other forcings in the CDR scenario are the same as in RCP45.

**3. Fast and slow components of radiation and precipitation**

Fast and slow components of radiation and precipitation were quantified from the component analysis simulations by regressing studied variable against temperature. These simulations were 20 year long. In each case, three ensemble members were simulated. Simulations were initiated in stable preindustrial conditions. In addition, studied forcing agent ($CO_2$ or SRM)





were included, which causes radiative imbalance and results in warming or cooling. Then, annual global mean values were

regressed against temperature to separate temperature-dependent and independent responses.

## 3.1 Evaluating the implementation of stratospheric sulfur aerosol fields in MPI-ESM and CESM

We evaluated the stratospheric aerosol implementation by comparing clear-sky aerosol radiative forcing in the two ESMs with that in ECHAM-HAMMOZ. The ECHAM-HAMMOZ simulations were performed with fixed sea surface temperatures, with aerosol radiative forcing calculated based on the change between a scenario with stratospheric sulfur injection and the control

simulation. To calculate the corresponding radiative forcing in the ESMs, a regression (Gregory) method was used (Gregory et al., 2004) (Fig. 2), which also provides the climate feedback parameter. First, we calculated the clear-sky shortwave flux and temperature anomaly compared to the Preind simulation for each year individually and made a linear regression between the two variables. Then, we obtained radiative forcing as the clear-sky shortwave flux anomaly of the linear regression line at zero temperature anomaly (i.e., when the climate system has not yet adjusted to the forcing).


The SW radiative forcing in both ESMs was in good agreement with that in ECHAM-HAMMOZ (dashed lines). Radiative forcings were slightly smaller in MPI-ESM than ECHAM-HAMMOZ, likely due to differing background conditions (preindustrial in MPI-ESM versus year-2000 in ECHAM-HAMMOZ, and thus more extensive ice cover in the MPI-ESM simulations). The zonal distribution of radiative forcing also agrees well between the models (not shown). Stratospheric

aerosols absorb some LW radiation, and the LW radiative forcing in MPI-ESM agrees well with that in ECHAM-HAMMOZ. However, CESM exhibits 37% (on average) weaker LW radiative forcing than ECHAM-HAMMOZ. This is probably due to the different radiative transfer models in CESM-CAM4 (9 LW radiation bands) and ECHAM-HAMMOZ (16 LW radiation bands). However, LW radiative forcing was small compared to the SW forcing, and this underestimation does not significantly affect the results or conclusions of this study. Because LW radiative forcing (warming effect) is weaker while SW radiative

forcing (cooling) is stronger in CESM than in MPI-ESM, SRM resulted in slightly more clear-sky cooling in CESM.

We see in Fig. 2 that SW radiative forcing does not increase linearly with the amount of injected sulfur. This is because more sulfur condenses onto existing particles, and small particles coagulate more efficiently with larger particles, when the sulfate burden is increased. This leads to lower particle numbers and larger particle sizes per unit sulfur injected (Heckendorn et al.,

2009; English et al., 2012; Niemeier and Timmreck, 2015). Conversely, Fig. 2 shows that the LW radiative forcing increased quite linearly with the amount of injected sulfur as shown, as also demonstrated by Niemeier and Timmreck (2015).

Earth's outgoing radiation is a linear function of temperature (Koll and Cronin 2018), an effect apparent in Fig. 2 c and d. However, SW radiation also changes as a function of temperature and we found that this change is fairly linear. The resulting

feedback was positive, amplifying cooling in the SRM scenarios and amplifying warming in the case of a $CO_2$ increase. The radiative fluxes in Fig. 2 are clear-sky, and this SW feedback is thus caused mainly by ice cover and albedo changes along





with changes in atmospheric absorption. The SW feedback was much larger in CESM (all-scenario average of 0.96 W/m2K) than in MPI-ESM (0.50 W/m2K). There was no large difference in surface albedo change between models. However, clear-sky SW absorption (net clear-sky SW flux at top of the atmosphere (TOA) - net clear-sky SW flux at surface) was linearly
dependent on surface temperature by 0.98 W/m2K in MPI-ESM and 0.85 W/m2K in CESM. We attribute this to the change in atmospheric water vapor due to the temperature change, which then contributes to atmospheric shortwave absorption. The differing model response likely originates from the distinct radiation schemes and spectral resolutions in MPI-ESM and CESM. This argument is supported by Fildier and Collins (2015), who likewise derived a larger SW absorption response to temperature in MPI-ESM compared to models that include CAM4.

Overall, we find that the clear-sky aerosol radiative forcings in the two ESMs are in good agreement with ECHAM-HAMMOZ. However, the same stratospheric sulfur fields yielded 8% weaker (on average) total (SW+LW) clear-sky radiative forcing in MPI-ESM than in CESM.

**3.2 Differences in Effective Radiative Forcings in MPI-ESM and CESM**

Figure 3 shows Gregory plots for the total TOA all-sky radiative forcing with clouds also taken into account. In this case, the total SRM radiative forcing was 22% weaker in MPI-ESM than in CESM. On the other hand, the radiative forcing due to increased $CO_2$ concentrations was larger in MPI (orange and red symbols in Fig. 3), but the difference was relatively small and is explained by different cloud forcings between models. The impact on SW radiation was larger than it was on LW radiation. The overall result is that the same stratospheric sulfur injection led to larger and faster cooling in CESM than in
MPI-ESM (Fig. 3). During the 20-year simulation period, stratospheric sulfur injections of 6 Tg(S)/yr (SRM6) led to slightly over -1 K global mean cooling (left-most green hexagon symbols in Fig. 3a) in MPI-ESM but closer to -2 K in CESM (Fig 3b). Global mean warming after 20-year $2xCO_2$ and $4xCO_2$ simulations was consistent between the models. However, there was a nearly 2 x larger radiative imbalance in MPI-ESM compared to CESM by the end of the simulations. If these simulations reached radiative equilibrium, the climate would presumably therefore be warmer in MPI-ESM than in CESM.

**3.3 Temperature-independent fast and temperature-dependent slow precipitation responses**

Precipitation responses can be divided into a temperature-independent fast response, which takes place immediately when some forcing agent is introduced, and a slow response caused by the temperature change and subsequent feedbacks (Myhre et al., 2017). Because of climate (e.g., ocean) inertia, precipitation will change slowly along with temperature even in the case of abrupt radiative forcing changes. Here, we separately quantified these fast and slow responses based on the regression method
described earlier. Results are shown in Fig. 4. Fast response was obtained by intersection of fitted line and the y-axes (T=0), and slope of the linear fit shows the slow response due to the temperature change. Fast responses are driven by changes in atmospheric absorption (Samset et al., 2016). A change in absorbed radiation modifies the amount of energy transferred between the TOA and surface. This energy transfer is then largely compensated by a change in latent heat flux (evaporation),



in turn changing precipitation. Changes in $CO_2$ concentration affect LW atmospheric absorption while SRM primarily modifies 290 SW reflection.

Figure 4 shows that an atmospheric $CO_2$ increase led immediately to a decrease in global mean precipitation. However, this $CO_2$ increase simultaneously warned the climate, which eventually led to a precipitation increase. After 2-5 years, this temperature-dependent slow component exceeded the immediate radiative component, and global mean precipitation was then 295 larger than in the absence of a $CO_2$ increase. On one hand, stratospheric sulfur aerosols (SRM1-6) also absorb some radiation (Fig. 2b), but on the other hand, relatively much more solar radiation is reflected and thus less is absorbed by background atmosphere. We therefore saw only a small total temperature-independent increase in global mean precipitation for most SRM cases. Overall increasing $CO_2$ decreases precipitation via the fast component and increases precipitation via the slow temperature component (Fig. 5). Fast precipitation impacts were significantly larger for $CO_2$ changes than for SRM (shown in 300 legends in Fig 4.), and therefore the fast precipitation component of SRM was omitted in Figure 5 for clarity.

As Fig. 4a shows, the fast precipitation responses in MPI-ESM differed from those in ECHAM-HAMMOZ, despite the fact that the same atmosphere model was used in both cases. This may result from differing background conditions between the models, land temperature change in ECHAM-HAMMOZ with fixed SST, or noise of in the yearly mean values of MPI-ESM 305 simulations.

Based on the scenarios examined here, the average global precipitation change scales with global mean temperature with a proportionality coefficient of 2.54 (SD 0.27) %/K in MPI-ESM and 2.26 (SD 0.13) %/K in CESM. These values are robust for temperature changes caused by $CO_2$ and SRM forcings. Our results thus support prior findings that the slow precipitation 310 response is not dependent on the forcing agent (Kvalevåg et al., 2013).

## 4. Results from simulated scenarios

In the scenario runs (Table 1), the years 2010-2100 were simulated for RCP45 and for geoengineering the RCP4.5 climate via SRM or CDR. Results are discussed below.

## 4.1 Change in global mean temperature

Global mean temperature and precipitation anomalies relative to 2010-2020 are shown in Fig. 1. Under RCP45, the global mean temperature increased by 1.30 K and 1.20 K in MPI-ESM and CESM, respectively. These changes were slightly below the CMIP5 multi-model mean of 1.35 K (Knutti and Sedláček, 2012). During the same period, global mean precipitation increased by 1.76-1.78 % under RCP4.5, also below the CMIP5 multi-model mean (2.66 %).



In the SRM-TEMP scenario, the global mean surface temperature was kept close to the present-day value via stratospheric sulfur injections. This reduced global mean precipitation in both ESM simulations (Fig. 6). The reduction was significantly larger in MPI-ESM (-1.42 %) than in CESM (-0.73 %). These differences are explored in Sect. 5. Given the SRM-TEMP results, it is not surprising that when global mean precipitation is maintained at the 2010-level in the SRM-PRECI scenario, the climate warms. SRM-PRECI warming in MPI-ESM is 0.64 K over 2010-2020, substantially larger than was seen in CESM

(0.27 K). This is consistent with the disparate model results for SRM-TEMP.

Overall, in both models, the majority of the global mean climate warming seen in RCP45 was compensated in SRM-PRECI. Based on GISTEMP data, the global average temperature in 2010-2018 was approximately 1 K warmer than in the pre-industrial era (defined as 1880-1900) (GISTEMP Team, 2019; Lennsen et al., 2019). Thus, in both ESMs the SRM-PRECI

global temperature increase stayed within the 2 C target of the Paris Agreement. For CESM, the SRM-PRECI temperature increase also stayed within the 1.5 C target.

The CDR scenario led to a 0.10 (MPI-ESM) and -0.11 (CESM) K change in global mean temperature by the end of the century (2080-2100) compared to present-day (2010-2020). There was thus no significant difference in global mean temperature

between the CDR and SRM-TEMP scenarios at the end of the century. The largest difference in global mean temperature between these scenarios was seen immediately after the onset of geoengineering, when the CDR temperature was larger than in SRM-TEMP. Under CDR, the global mean temperature only starts to decrease post-2040. This is because CDR acts more slowly to reduce global temperatures than does stratospheric sulfur injection (Royal Society 2008). In the CDR scenario, $CO_2$ removal was suspended in year-2070, when atmospheric $CO_2$ concentrations have returned to their 1976 levels. The global

mean temperature at that time was close to the present-day value, and did not change significantly through the end of the century (when the rate of change in atmospheric $CO_2$ matches that seen in RCP 4.5). Thus, even this very optimistic CDR scenario is insufficient for cooling climate to pre-21st century levels. However, our CDR scenario only reduced $CO_2$ concentrations, with other GHGs and aerosol concentrations following RCP 4.5.

**4.2 Change in global mean precipitation**

Although the global mean surface temperature in the CDR scenario was the same at the end of the century (2080-2100) as at the beginning of the century (2010-2020), the global mean precipitation was over 0.5 % larger in both ESMs. In Sect. 3.3, we showed that the precipitation impacts of SRM and $CO_2$ can be separated into a temperature-independent fast component and a temperature-dependent slow component. Here we use that framework to examine precipitation responses across the different geoengineering scenarios. Precipitation is also affected by non- $CO_2$ GHGs, tropospheric aerosols, and land-use changes, all

of which can induce their own temperature-independent fast components. For our purposes this can be assumed to be the same across all scenarios.



We thus describe the global mean precipitation change as the sum of the temperature dependent slow component (a*ΔT) and all fast components (Fläschner et al., 2016):

$$\Delta P = a \times a\Delta T + b(SRM) + c \times \ln \frac{CO_{2\,preind} + \Delta CO_2}{CO_{2\,preind}} + BG, \tag{1}$$

where a,b,c are model-specific coefficients, T is simulated global mean surface temperature, $CO_{2\,preind}$ is the preindustrial $CO_2$ concentration, $\Delta CO_2$ is the atmospheric $CO_2$ change relative to the preindustrial value, and BG is the background fast component, assumed to be the same for all scenarios. Coefficient a is obtained from the scenario-ensemble mean slope in Fig 4, while b is the fast component (intercept) from simulations of the corresponding SRM scenario. The fast precipitation
response varies linearly with absorbed radiation (Myhre et al., 2017). Radiative forcing due to $CO_2$ varies logarithmically with concentration (Etminan et al., 2016). Thus, the fast precipitation response for $CO_2$ is logarithmically dependent on $CO_2$ concentrations. We calculated the fast precipitation response for three different $CO_2$ concentrations: preindustrial, 2x$CO_2$, and 4x$CO_2$. The coefficient c can then be calculated from a logarithmical fit of the fast response versus $CO_2$ concentration across these three scenarios. This approach yields c values of 4.5 (%) for MPI-ESM and 4.0 (%) for CESM. Finally, we calculated
the BG component as the 5-year running mean residual between the first three right-hand terms of Eq. (1) and the modelled precipitation in the RCP45 scenario. Note that if Eq. (1) is used only to study precipitation change between scenarios, the BG component is not needed (see supplementary material Fig. S1). However here we also wish to examine precipitation changes relative to 2010-2020, and the BG term is thus included here.

Figure 7 shows the precipitation component for each scenario in MPI-ESM and CESM. In general, the precipitation signal as estimated by the fast and slow components via Eq. (1) corresponds well to the actual model quantity in both ESMs for all scenarios. From year 2020 to 2100 mean error in results between Eq. (1) and actual model quantity were 0.01%, 0.04%, and -0.01% with MPI-ESM and -0.16%, -0.12%, and 0.05% with CESM in SRM-TEMP, SRM-PRECI, and CDR, respectively. These individual components can therefore be used to understand the drivers of precipitation change for each scenario.
Figures 7a and b show that the precipitation increase in RCP45 would be roughly twice as large if only the slow component were operative. However, the fast radiative component reduced global mean precipitation increase by over 1% in both ESMs. This fast component related to increasing atmospheric $CO_2$ (plus other GHG and absorbing aerosol) probably also explains why the increase of observed global mean precipitation to date has not increased significantly to date, despite the fact that
climate has warmed (Allan et al., 2014).

Under the SRM-TEMP scenario (Fig. 7 c and d), the global temperature change (and thus the slow precipitation component) was small, as is the fast precipitation component due to sulfate aerosols. However, the fast component due to $CO_2$ was as large as in RCP45. This fast (radiative) component from $CO_2$ is the main reason that SRM generally leads to a decrease in global

mean precipitation when used to fully offset GHG-induced warming. On the other hand, in the SRM-PRECI scenario (Fig. 7
       e and f) the climate was cooled to the point that the temperature-dependent slow component balances the fast radiative
       components ($CO_2$, SRM and background), so that the net precipitation change was close to zero.

       The CDR scenario led to a slight increase in global mean precipitation, despite no significant net change in global mean
temperature. Figures 7 g and f show that this was also explained by the fast radiative component of $CO_2$. As in SRM-TEMP
       the slow temperature-dependent component was small. However, atmospheric $CO_2$ concentrations were much lower by the
       end of the century, reducing atmospheric absorption and thus increasing global precipitation compared to 2010-2020.

       Although the global mean precipitation response was approximately the same in both ESMs in RCP45 and CDR, a closer look
at the underlying drivers shows that only the radiative component of $CO_2$ was consistent across models. The temperature-
       dependent response differs between ESMs, driving divergent precipitation impacts. This was resulted from slightly different
       temperature response and hydrological sensitivity between ESMs. In RCP4.5 temperature-dependent slow component was
       32% larger in end of the simulation (2080-2100) with MPI-ESM than in CESM simulation. In CDR magnitude of slow
       component was same between models (0.28% in MPI-ESM and -0.24% in CESM at the end of the simulation), but a sign was
different. However, this effect was compensated by differing non-$CO_2$ background responses, which also changed over the
       course of the simulated century. Figure 7 shows that this BG response is very different between the models and even have a
       different sign. In MPI-ESM non-$CO_2$ fast components were causing 0.48% decrease on precipitation at the end of the
       simulation (2080-2100) compared to the beginning (2010-2020) while in CESM non-$CO_2$ forcers were increasing precipitation
       by 0.23% Thus, it is merely fortuitous that the net precipitation response was similar between models in the CDR and RCP45
scenarios.

       The BG radiative components impacting precipitation include a range of factors including non-$CO_2$ GHG (methane, nitrous
       oxide, ozone, CFCs etc), tropospheric and background stratospheric aerosols, and land-use change—with differing treatments
       between models. Radiative transfer modeling also differs between the ESMs. As shown in Sect. 3.1-3, radiative forcing and
(particularly) atmospheric absorption—and thus latent heat flux and precipitation—in the ESMs responded differently to the
       various forcing agents. Thus, it is not surprising that the BG precipitation component, which is affected by several different
       forcing agents, also differs between models.

## 5. Sulfur injections

       Figure 8 shows the amount of sulfur required to keep global temperature or precipitation at current levels through the end of
the 21st century. All scenarios started with injections of 1 Tg(S)/yr in year-2020, and the amount of required sulfur then
       increased along with the RCP4.5-driven warming. In all cases, more sulfur was needed to compensate for RCP4.5 warming

than for the associated precipitation increase (see cumulative injection amount on right-hand axes). As shown in Sect. 4.2, the fast, $CO_2$-driven radiative component partly offsets the temperature-driven precipitation component caused by global warming. Thus, in the SRM-PRECI scenario, the sulfur aerosol only needs to compensate for the (already partly offset) precipitation effect of changing temperatures, rather than for the total temperature change (as is the case in SRM-TEMP).


Based on these simulations, in a total of 107-113 Tg(S) and 95-114 Tg(S) was required to prevent a simulated precipitation increase between years 2020 and 2100 in MPI-ESM and CESM, respectively (scenario SRM-PRECI). These 80-year totals are slightly larger than the amount of $SO_2$ emitted each year in the mid 1970s, when annual emissions were roughly 75 Tg(S)/yr (Smith et al., 2011). Global sulfur emissions have since decreased; however, China alone emitted over 100 Tg(S) $SO_2$ between 2006 and 2008 (Li et al., 2017). However, the lifetime of aerosols derived from surface emissions is on the order of days, and the cooling impact is therefore much smaller than in the case of stratospheric injection. In the SRM-PRECI scenario, yearly injections are 3 Tg(S)/yr or less, with the exception of occasionally higher injections for one MPI-ESM ensemble member.



Figure 8 reveals significant differences in injection amounts between the two ESMs. In CESM, preventing global mean warming (under RCP4.5) through year-2100 requires a total of 163-199 Tg(S). This was less than twice the amount required to prevent an increase in global precipitation. However, simulations with MPI-ESM suggest that preventing global mean warming via SRM would require 292-318 Tg(S), 50-100% more than in CESM and approximately 3x the amount required to stabilize global mean precipitation in scenario SRM-PRECI. Maximum yearly injections reached 6 Tg(S)/yr in MPI-ESM but only 3 Tg(S)/yr in CESM. These differences are mostly explained by the model responses to sulfate aerosols shown in Sect. 3: the all-sky forcing for a given amount of sulfur was significantly (22-33 %) larger in CESM than in MPI-ESM.


Figure 8 also shows some limitations of the climate-control algorithm used here. At times the change in SRM injection amount was too large, leading to an overly-strong climate response. In some cases the ensuing compensatory change in injection amount then overshoots the desired climate response in the opposite direction. This led to rapid fluctuations between SRM levels, as seen for example between years 2070-2080 in MPI-ESM ensemble member 2 for the SRM-PRECI scenario. Such effects could be avoided by smaller injection increments or by using a more sophisticated algorithm that could better separate large natural variations in temperature or precipitation from long-term changes. We also noted that the introduction of 1 Tg(S)/yr in 2020 led to an overly-large precipitation response for all simulations under scenario SRM-PRECI. However, the above effects do not affect the overall results and conclusions shown here.



## 6. Regional climate responses

While the SRM-TEMP and CDR scenario simulations led to similar global mean temperatures by the end of 21st century, the regional responses were quite different. Figures 9 a and b map the temperature difference between these two scenarios in both

ESMs for the last 20 years of the 21st century. We see that the SRM-TEMP scenario led to cooler tropics and warmer high
latitudes than the CDR scenario in both ESMs. These regional discrepancies have been demonstrated in prior studies (Kravitz
et al., 2013; Laakso et al., 2017) and point to a fundamental problem with the SRM approach. Aerosols primarily affect
incoming SW radiation, while GHGs affect LW thermal radiation, and the meridional gradient is steeper for SW than for LW
radiation. Consequently, compensating for a global mean LW change by modifying SW radiation leads to zonally-dependent
differences. This issue can be reduced by concentrating the SRM injections in mid and high latitudes, or via seasonal
adjustment of the sulfur injection area (Laakso et al., 2017). Overall, however, the temperature differences over land between
scenarios were rarely statistically significant (indicated by hatching in Fig. 9).

Figures 9 c and d compare the SRM-TEMP scenario to present-day climate (2010-2020). In MPI-ESM, the regional SRM-
TEMP versus present-day temperature differences were significantly larger than those between SRM-TEMP and CDR at the
end of the century. However, this was not the case in the CESM simulations. It should be kept in mind that comparing years
2080-2100 from the SRM-TEMP scenario with 2010-2020 (as present-day) is a somewhat arbitrary choice, and that the
comparison reflects not only geoengineering impacts but also climate change under RCP4.5. In addition, even though the
global mean temperature was similar between these two periods, the climate was relatively stable in 2080-2100 but was
warming in 2010-2020 climate. The regional patterns seen in Fig. 9 c) and d) thus depend to a degree on the choice of reference
years, and not only the impacts of geoengineering.

Regional temperature anomalies for other scenarios are provided in the supplement (Fig. S2). Overall, RCP45 led to larger
warming at high latitudes than at low latitudes when compared to CDR for years 2080-2100. The corresponding regional
patterns in SRM-PRECI were similar to those in RCP45 but with reduced magnitude. Nevertheless, warming in SRM-PRECI
relative to the CDR scenario was statistically significant almost everywhere in both models.

Figure 10 shows the relative precipitation differences between the SRM-PRECI and CDR scenarios in boreal winter (DJF) and
summer (JJA) in 2080-2100. Globally, CDR led to 0.5% more precipitation than SRM-PRECI in both models. However, this
precipitation change was not regionally or seasonally homogeneous. A key conclusion is that these changes were rarely
statistically significant (hatching in Fig. 10) and that there was often not good agreement between models.

Both models did show broadly similar responses over tropical oceans, especially over the Eastern Pacific and Atlantic. This
was probably caused by an Intertropical Convergence Zone (ITCZ) shift due to the zonal temperature difference between
SRM-PRECI and CDR (SRM-PRECI led to more warming in high versus low latitudes compared to CDR). Generally, the
responses seen in Fig. 10 were larger in MPI-ESM than in CESM, likely due to the significantly warmer climate in MPI-ESM
under SRM-PRECI. Supplemental Figures S3 and S4 show that when comparing temperature in SRM-PRECI and CDR,
simulations with MPI-ESM led to much greater warming in DJF and (especially) JJA over Europe, Australia and South

America when compared to CESM. Figure 10 shows that the corresponding precipitation responses were also significantly different over these areas. Precipitation responses for the other studied scenarios are shown in the supplement (Figs. S5 and

S6). As with the results in Fig. 10, spatial features of these precipitation responses were rarely statistically significant. To increase confidence in how SRM and CDR would affect regional precipitation distributions, longer simulations or larger ensemble sizes are necessary.

## 7. Discussion and conclusions

Here, we have studied different scenarios in which global mean warming and precipitation changes are compensated by solar

radiation management (SRM) or carbon dioxide removal (CDR) during the 21st century. We carried out simulations using two Earth System Models, MPI-ESM and CESM, with SRM based on stratospheric aerosols first simulated with the aerosol-climate model ECHAM-HAMMOZ. SRM was used for two scenarios in which the magnitude of sulfur injections was controlled to maintain global mean temperature or precipitation at year 2010-2020 levels in the RCP4.5 scenario. Additionally, an idealized CDR scenario (also based on RCP4.5) was performed that included 1%/yr removal of atmospheric $CO_2$. We

examined the resulting global mean precipitation changes mechanistically by dividing the response into temperature-dependent and temperature-independent components. These model-specific components were defined based on a regression method using simulations with fixed climate conditions, and that included a constant SRM treatment or an abrupt change in atmospheric $CO_2$ concentrations.

Our work supports prior studies in showing that the ratio of the global precipitation change to the global temperature change for SRM is larger than for an atmospheric $CO_2$ perturbation (e.g. Bala et al., 2008). Thus, less sulfur was needed to compensate for the global mean precipitation change under RCP45 than to compensate for the corresponding temperature. Our results showed that maintaining global precipitation at the same level from 2010 to 2100 required a total of 107-113 Tg(S) (with MPI-ESM) and 95-114 Tg(S) (with CESM). However, preventing an increase in global mean temperature required 292-318 Tg(S)

(with MPI-ESM) and 163-199 Tg(S) (with CESM). Keeping global precipitation at current levels through 2100 would thus require roughly the same amount of sulfur as the estimated surface emissions of China alone between 1996-2005 (121 Tg(S), Smith et al., 2011) http://sedac.ciesin.columbia.edu/data/set/haso2-anthro-sulfur-dioxide-emissions-1850-2005-v2-86/data-download). This simultaneously reduced global mean warming by 50% and 78% based on the MPI-ESM and CESM simulations, respectively (compared to the 2010-2100 RCP4.5 temperature increase in the absence of SRM).


While completely preventing global mean warming in this century (in RCP4.5) would require much more sulfur than preventing a change in global precipitation, the total sulfur required was comparable to that emitted globally at the surface from anthropogenic sources during the first five years of the 21st century (274 Tg(S), Smith et al., 2011). However, maintaining a constant global mean temperature in this way led to a significant reduction in global mean precipitation (-1.42% with MPI-

ESM and -0.73% with CESM) compared to present-day climate. Our component analysis showed that this precipitation decrease was caused by the temperature-independent radiation component resulting from the $CO_2$ increase in the RCP 4.5 scenario. Under RCP45 without SRM, this component was overridden by the temperature-dependent effect on precipitation from global warming. When this temperature component was compensated by SRM, the $CO_2$ component remains and global mean precipitation decreases. It should be noted that this is the case for all SRM methods and not only for stratospheric

aerosols. SRM itself had only a small temperature-independent fast effect on precipitation.

In the CDR scenario, the annual $CO_2$ increase based on RCP4.5 was counteracted by a 1% annual removal of the atmospheric $CO_2$ concentration until year 2070. This was found to slow down warming significantly and to return the global mean temperature to its present-day (2010-2020) value. The atmospheric $CO_2$ budget is currently increasing at roughly 4 GtC/yr. In

our CDR scenario, 8.7 GtC of $CO_2$ was removed yearly after year-2020. Our scenario should be considered as an idealized high-end CDR scenario as achieving this high $CO_2$ removal rates in a few years would not be feasible due to technological, economic, social, and political issues. The results highlight the challenge in substantially slowing global warming, and suggest that entirely preventing global-mean warming during this century solely via CDR without significant cuts in $CO_2$ emissions is probably not achievable.


Even though global mean temperature at end of the CDR simulation was the same as at the beginning, global mean precipitation increased (~0.5 %) in both ESMs. To date, we have not seen as large increase in global mean precipitation as would be expected only based on the temperature increase (Allan et al., 2014). This is because the fast radiation-driven precipitation effect is largely compensating the slower temperature-dependent component from warming. However, over time, the temperature

component will dominate, and a significant increase in global mean precipitation is expected. If atmospheric $CO_2$ is removed as in the CDR scenario, the temperature component is prevented from increasing, but simultaneously a positive fast $CO_2$ precipitation component is induced by the reduction of $CO_2$, increasing global mean precipitation. It is thus difficult to prevent an increase in global mean precipitation via GHG reduction. However, global precipitation changes are also driven by the fast radiative components of aerosols and non-$CO_2$ GHGs, and future precipitation will depend on how these emissions evolve

over time.

RCP45 and CDR scenarios led to a similar global mean precipitation response between the two ESMs. However, regression analysis revealed that this was fortuitous. The precipitation response to changing temperature and $CO_2$ concentrations differed between the ESMs, but these differences were masked by offsetting background (BG) effects related to other GHGs and

tropospheric aerosols. Large differences in the primary drivers of precipitation change can therefore exist between ESMs even when the ESMs predict similar net changes. A more detailed component analysis, with BG effects separated into relevant subcomponents, is therefore needed. The Precipitation Driver Response Model Intercomparison Project (PDRMIP) may help address this issue (Myhre et al., 2017).

Similar component analyses as done here on the global scale (Sect. 4.2) can in principle be performed regionally. However, for regional analyses (e.g., applying Eq. (1) for a single model grid box), the dry static energy flux divergence of the atmosphere needs to be taken into account (Richardson et al., 2016). This term depends on the neighbouring grid boxes and is not linear or independent from other components. Because of this and natural variability, regression analyses to quantify the fast and slow precipitation components either regionally or for individual grid boxes will be subject to noisier data than in the global

case. However, preliminary analyses reveal regions where the approach appears promising, and we therefore recommend further evaluation of this potential in subsequent work.

Overall, this study shows that global mean temperature-independent fast and temperature-dependent slow precipitation responses caused by CDR and SRM can be quantified by the regression method. When these components are known, the

global mean precipitation change can be presented as the sum of the temperature dependent slow component and all fast components. Our results show that the fast responses of $CO_2$ have a major role in the resulting precipitation impacts, when $CO_2$ induced global warming is slowed down by geoengineering. If global warming is prevented by stratospheric sulfur injections while atmospheric $CO_2$ concentration still increases, the global mean precipitation is decreased due to the fast response of increasing atmospheric $CO_2$. On the other hand, less sulfur is required to keep the global mean precipitation stable,

because the fast precipitation response to increased $CO_2$ is the opposite of the slow precipitation response resulting from warmer climate. Without SRM, temperature response overruns $CO_2$ fast response (as in RCP45). Also in our CDR scenario, the global mean precipitation increase was explained by the positive fast precipitation response to reduced $CO_2$. As we showed here, separating precipitation to fast and slow response is a useful method to analyse differing precipitation responses between different geoengineering techniques. This framework can thus help to understand and anticipate temperature and precipitation

responses in different time scales and geoengineering scenarios, where SRM and CDR are used potentially simultaneously. In principle, this method can also be used to study precipitation response in any scenario, if the temperature change and forcing agents are known.

*Code and data availability.* The data from the model simulations and implemented model codes are available from the authors

upon request.

*Author contributions.* AL designed the research, performed the experiments, carried out the analysis and prepared manuscript. All author contributed ideas, interpretation and discussion of results and to writing the paper.

*Competing interests.* The authors declare that they have no conflict of interest.



Acknowledgements. A. Laakso was supported by Tiina and Antti Herlin Foundation. A.-I. Partanen was supported by the Fonds de Recherche du Québec - Nature et technologies (grant number: 200414), Concordia Institute for Water, Energy and Sustainable Systems (CIWESS) and the Academy of Finland (grant number: 308365). The ECHAM–HAMMOZ model is
developed by a consortium composed of ETHZ, Max-Planck Institut für Meteorologie, Forschungszentrum Jülich, University of Oxford and the Finnish Meteorological Institute and managed by the Center of Climate Systems Modeling (C2SM) at ETHZ.

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

| Component analysis simulations (Preindustrial) | 20 years with fixed background conditions |
|---|---|
| Preind | Fixed preindustrial conditions |
| 2xCO$_2$ | Atmospheric $CO_2$ concentration = 570 pm |





| 4xCO$_2$ | Atmospheric CO$_2$ concentration = 1140 pm |
|---|---|
| SRM1-6 | Continuous 1-6 Tg(S)/yr sulfur injections (10 N - 10 S, 20km) |
| | |
| **Scenarios** (RCP 4.5) | years: 2010 - 2100 with RCP 4.5 scenario in background |
| RCP45 | Representative Concentration Pathway 4.5 W/m$^2$ |
| SRM-TEMP | Temperature kept at 2010-2020 level by SRM |
| SRM-PRECI | Precipitation kept at 2010-2020 level by SRM |
| CDR | In addition to yearly change in atmospheric CO$_2$ concentration in RCP 4.5, 1%/yr CO$_2$ is removed from atmosphere |

**Table 1. Simulations**

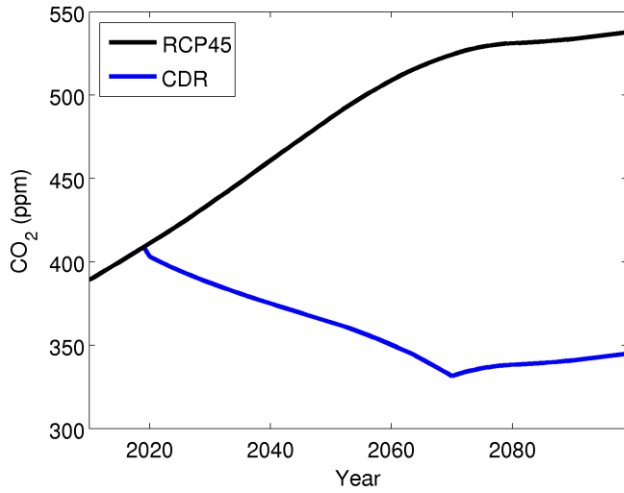

**Figure 1: Atmospheric carbon dioxide concentration in scenarios RCP45 and CDR.**




**Figure 2: Gregory plots of the shortwave radiative flux change (clear-sky conditions) with a) MPI-ESM and b) CESM, and of the longwave radiative flux change (clear-sky conditions) with c) MPI-ESM and d) CESM. Markers indicate a single-year global mean value in one ensemble member and solid lines are linear fit lines. Dashed lines show aerosol clear-sky radiative forcing in ECHAM-HAMMOZ, with numerical values shown in the middle. Corresponding radiative forcing (intersection of linear fit and the y-axes (T=0)) in MPI-ESM and in CESM are shown in legends next to the figure. Origin represents zero temperature and clear-sky radiative flux anomaly compared to Preind simulation.**




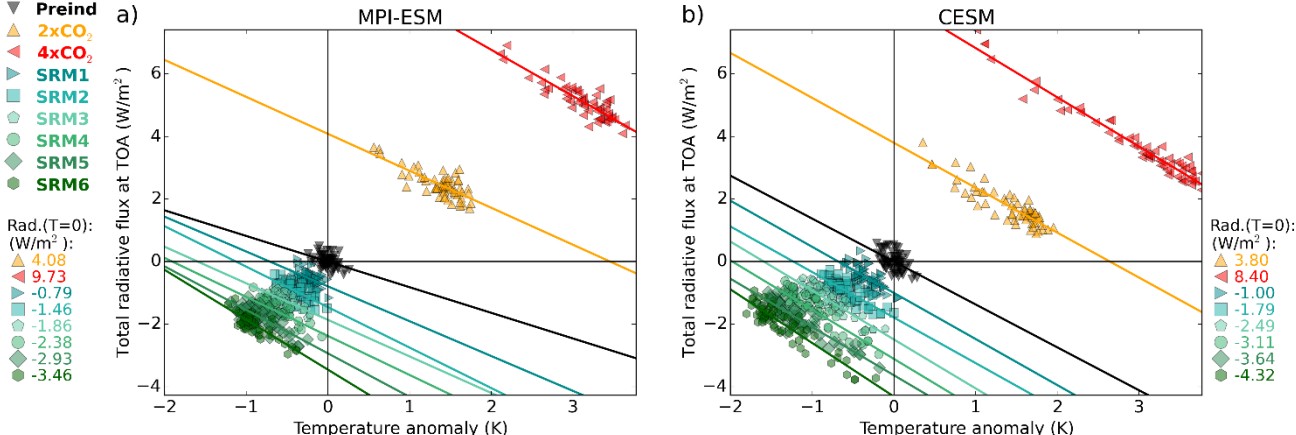

**Figure 3: Gregory plots of total all-sky radiative flux change at the top of the atmosphere for a) MPI-ESM and b) CESM. Markers indicate a single-year global mean value for one ensemble member and solid lines are linear fits. Corresponding all-sky radiative forcing (intersection of linear fit and the y-axes (T=0)) in MPI-ESM and in CESM are shown in legends next to the figure. Origin represents zero temperature and clear-sky radiative flux anomaly compared to Preind simulation.**

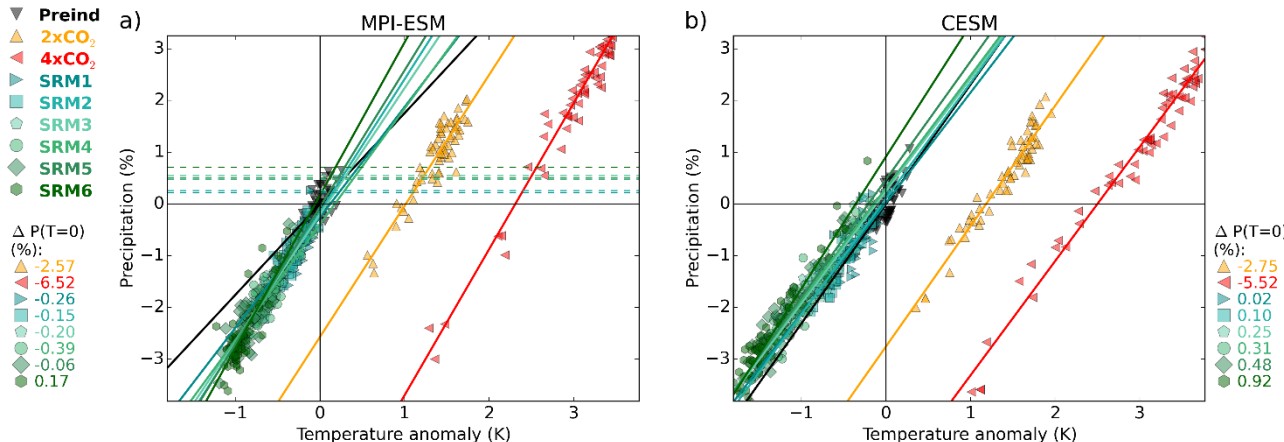

**Figure 4: Gregory plots of global precipitation changes under increased $CO_2$ (orange and red) and SRM scenarios with differing sulfur injection amounts (blue to green) for a) MPI-ESM and b) CESM. Each markers indicates a single-year global mean value for one of three ensemble members and solid lines are linear fits. Origin represents zero temperature and precipitation anomaly compared to Preind simulation. Fast precipitation response is obtained from intersection of linear fit and the y-axes (T=0) (shown in legends next to the figure), and slope of the linear fit corresponds the slow response due to the temperature change. Dashed lines show (fast) precipitation responses for the corresponding scenarios in ECHAM-HAMMOZ (simulations with fixed SST).**



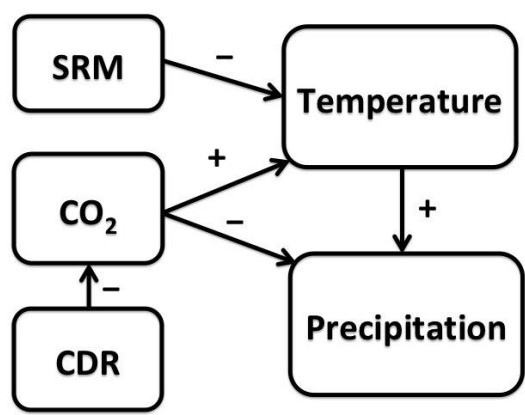

**Figure 5: A schematic presentation of fast radiatively-induced and slow temperature-induced components of SRM and CDR. Plus and minus signs indicate the direction of change in the target variable when the driving variable is increasing. If the driving variable is decreasing (e.g., temperature decrease due to SRM), the target variable changes in the opposite direction as indicated (e.g., decrease in precipitation due to decreased temperature). The fast component of SRM is so small compared to that induced by changes in CO₂ concentration that it is omitted for clarity.**

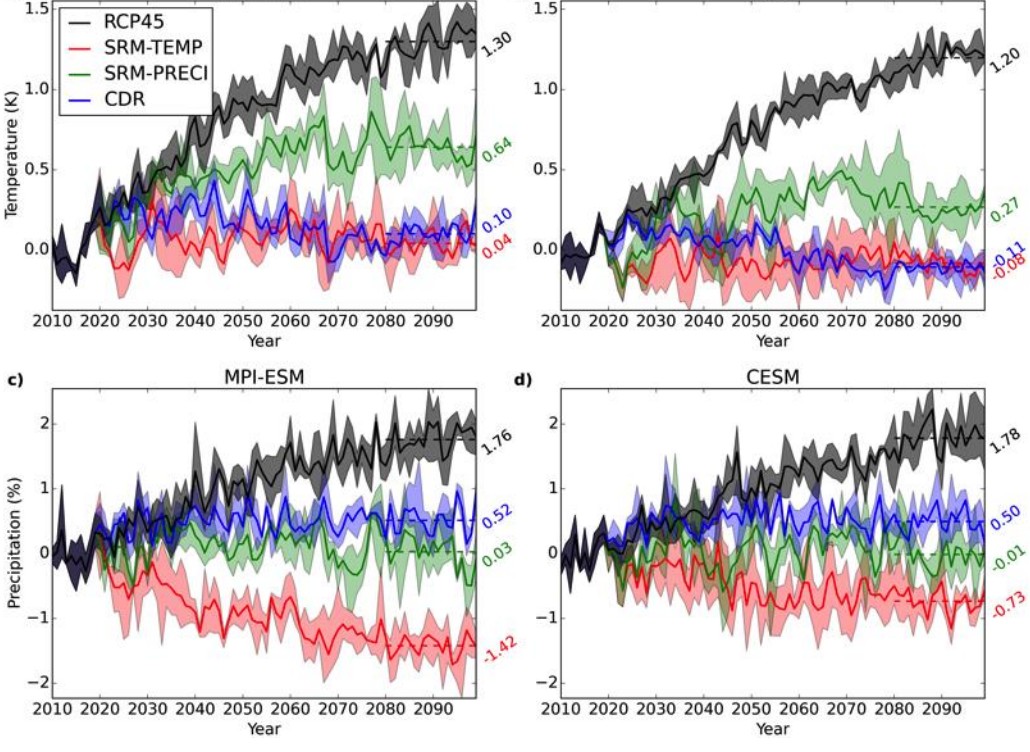

**Figure 6: Global mean temperature anomalies in a) MPI-ESM and b) CESM, and global mean precipitation anomalies in c) MPI-ESM and d) CESM. Numbers to the right of each figure indicate the global mean difference between 2080-2100 and 2010-2020. Shaded areas show the maximum and minimum across three ensemble members.**



**Figure 7: Precipitation components for each of the simulated scenarios. Solid colored lines with shaded areas have the same meaning as in Figures 5 c and d. Dashed colored lines indicate the precipitation change caused by individual component (see legend in panel b) for each scenario and model. The purple solid line shows the sum of all precipitation components (T, SRM, CO₂, and BG).**





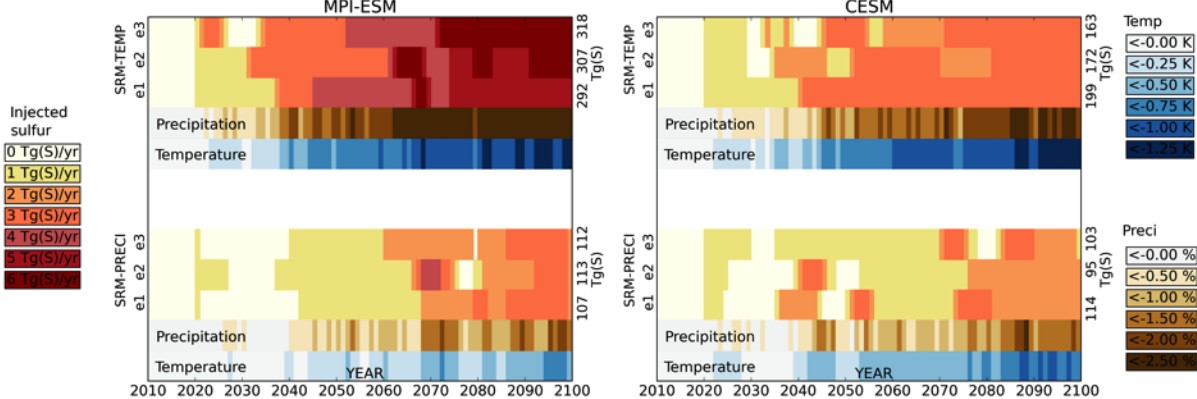

**Figure 8: Yearly sulfur injections in scenarios SRM-TEMP and SRM-PRECI for three ensemble members in MPI-ESM (left) and CESM (right). Also shown are the corresponding global mean precipitation and temperature differences relative to RCP45. The cumulative injection amount for each ensemble member is listed on the right-hand axis.**

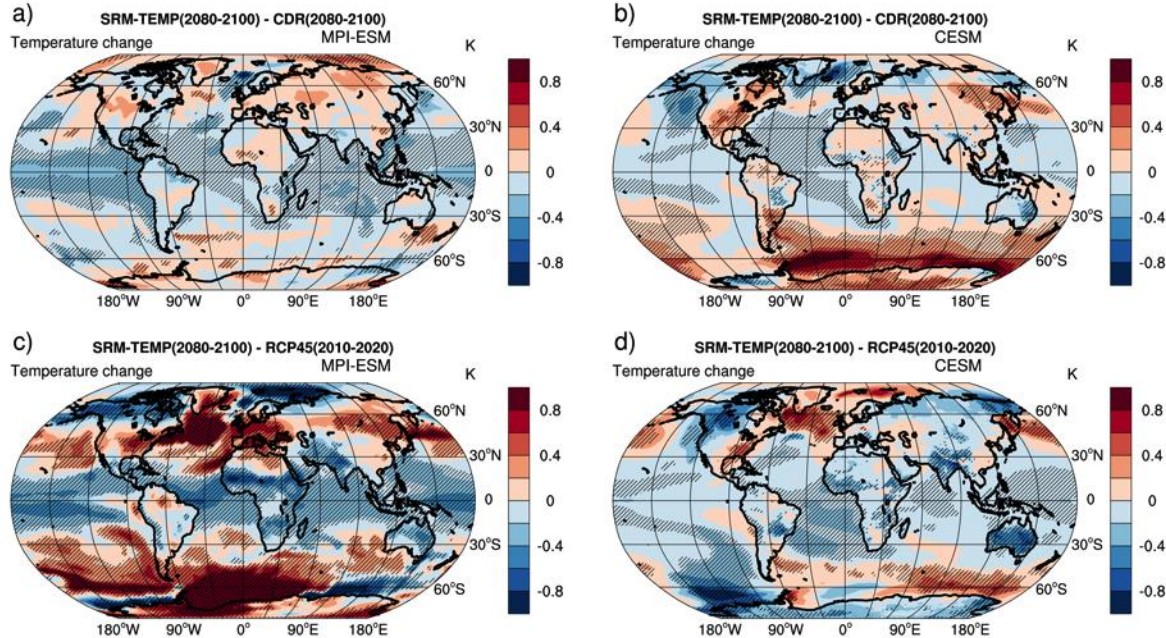

**Figure 9: Differences in regional temperature patterns between the SRM-TEMP and CDR scenarios for years 2080-2100 in a) MPI-ESM and b) CESM. Also shown are the temperature differences between the SRM-TEMP scenario for years 2080-2100 and present-day climate (RCP45, years 2010-2020), in c) MPI-ESM and d) CESM. Hatching indicates regions where the temperature change is statistically significant at the 95% level, with significance levels estimated using a Student's paired t-test (sample of 20 yearly mean values for 3 ensemble members).**



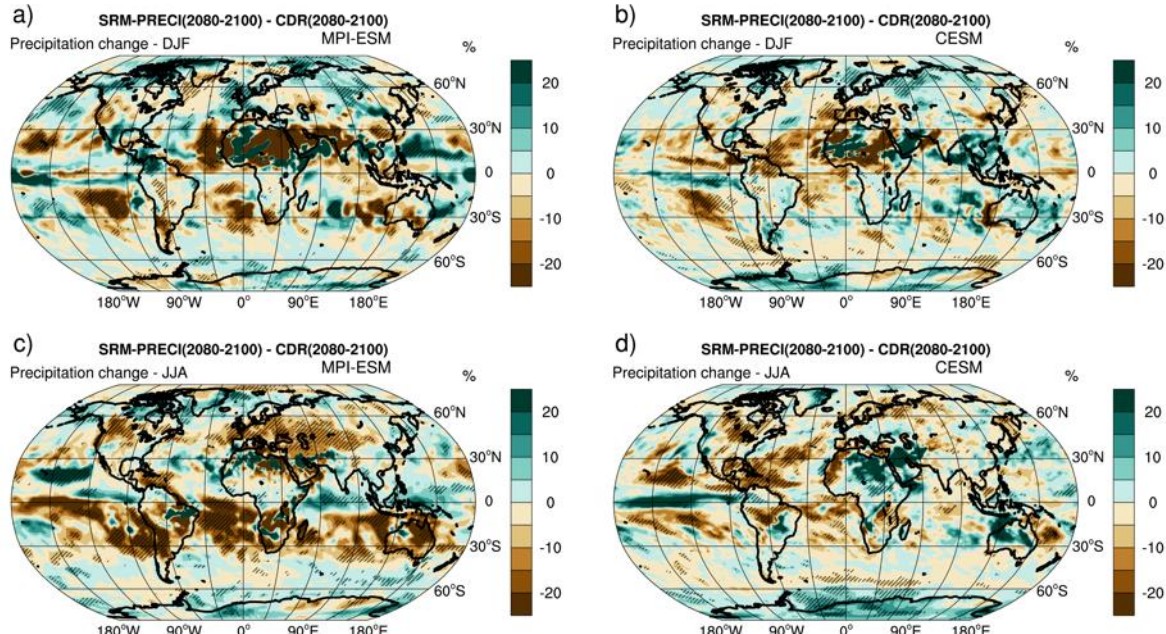

**Figure 10: Relative change in precipitation between the SRM-PRECI and CDR scenarios for a) December-January-February and c) June-July-August in MPI-ESM, along with the corresponding figures for CESM (b and d). Hatching indicates regions where the temperature change is statistically significant at the 95% level, with significance levels estimated using a Student's paired t-test with (sample of 20 yearly mean values for 3 ensemble members).**

865