# Peer review of "Differing precipitation response between Solar Radiation Management and Carbon Dioxide Removal due to fast and slow components"

_Earth System Dynamics, 2019_

## Referee Comment (RC1) · Anonymous Referee #1 · 17 Nov 2019

General comments

The authors studied the climate responses to two different geoengineering approaches: sulfur injections and carbon dioxide removal. Using the Gregory regression approach, the authors separated temperature-independent (fast adjustment) and temperature-dependent (slow responses) components and then quantified their contributions to the total precipitation change in an RCP based scenario. The authors also compared the total amount of aerosols needed under various scenarios and the spatial pattern changes. The paper is well written and contributes to our understandings of the hydrological cycle responses to different geoengineering forcing. I would recommend

publication after the authors addressed the following comments.

Specific comments

My understanding is that during the simulations using MPI-ESM and CESM, you directly used the "equilibrium" aerosol distributions patterns estimated from ECHAM-HAMMOZ under the 1, 2, 3, 4, 5, and 6 Tg(S)/yr emission rates (in Fig.8 you used discrete injected sulfur rates). In the real world, if you change the emission rate of aerosols in year N, it takes time to adjust to a new equilibrium state. Therefore, there is a difference between your simulations and the real-world implementations (maybe it's worth to point out?). Do you have any idea how much the difference could be? The discussion of regression in Section 3 is good. But in general, I think that the authors could add more information to support their statements and help the reader to better understand. For example, it would be very helpful if the author could show numbers at least for those regression slopes. It's hard to identify and compare these regression lines with statements in the paper by eyes. Another example is shown in lines 258 to 260: the authors stated that "There was no large difference in surface albedo change between models" and "clear-sky SW absorption was linearly dependent on surface temperature". Can you remind me where I can find figures or data for such statements? For each of the two models, the authors used regression results from three 20-year simulations (preindustrial, 2xCO2, and 4xCO2) and applied a logarithmical fit to estimate the coefficient "c" (the fast CO2 effect to precipitation) (lines 362 to 364). You have only three data points during the curve fit, so I am wondering how good the curve fit is? The fast precipitation response to CO2 change not only relates to the CO2 radiative effect but is also partly attributed to the physiological effect. It's fine to use the logarithmical form (I think the first paper used this relationship is in Cao et al., (2015)), but I think you might want to point it out here.

Technical corrections

Line 34: Add "increase" to "maintain the global mean temperature increase within 2°C

[Figure]

. . ."

Line 39: When the author says "a reduction of energy use", do you mean energy use associated with fossil fuel burning only? Because energy consumption provided by renewables and biofuels will not deteriorate the climate warming target.

Line 47: Missing ")" here: "(Luderer et al., 2018, IPCC 2018)"

Line 67: ". . . SRM would decrease global mean precipitation through the direct radiative effect described above". The hydrological cycle responses to solar geoengineering depend on magnitudes of geoengineering deployments and which case you are comparing with (the pre-industrial or high-CO2 world without SRM). I think the statement here should be more specific.

Line 80: add "for" to: ". . . continue to change for decades. . ."

Line 89: "separate" to "separates", "totally" to "total"

Line 94: "a" to "the"

Line 134: delete "the"

Line 182: what do you mean of "business as usual scenarios" here?

Line 196: add "into" to "taken into account"

Line 217: add "as"

Line 224: "were" to "was"

Line 232: Please define "Preind" before using it

Line 293: "warned" to "warm"

Line 294 and 325: the font is different

Line 315: "Fig.1" should be "Fig.6"

[Figure]

Line 324: "2010-2020" should be "2080-2100"

Line 355: there is an additional "a" in Eq. (1)

Reference

Cao, L., Bala, G., Zheng, M., & Caldeira, K. (2015). Fast and slow climate responses to $CO_2$ and solar forcing: A linear multivariate regression model characterizing transient climate change. Journal of Geophysical Research: Atmospheres, 120(23), 12–37.

———————————————————

---

## Referee Comment (RC2) · Tamas Bodai (Referee) · 12 Dec 2019

The authors model two geoengineering methods, the Solar Radiation Management (SRM) and Carbon Dioxide Removal (CDR), in two Earth system models, the CESM and the MPI-ESM, and run simulations under several scenarios and analyse the global mean and regional temperature and precipitation responses. The considered scenarios are the following:

(1) RCP4.5

(2) RCP4.5 with SRM controlled to approximately maintain present-day global mean

temperature throughout the 21st c.

(3) RCP4.5 with SRM controlled to approximately maintain present-day global mean precipitation throughout the 21st c.

(4) RCP4.5 with an ambitious CDR removing 1% of $CO_2$ concentration per year

They give some evidence in both ESMs that the global mean precipitation response has a global mean temperature-dependent component, which is based on the methodology proposed by Gregory et al. (2004). Besides this the precipitation can respond very fast, at least to the greenhouse $co_2$ forcing, and this fast response is of opposite sign compared with the temperature-dependent part. To aerosol forcing the fast response is negligible. This is the reason why there is less precipitation in an SRM scenario when global warming is fully compensated. Conversely, the authors make the point and demonstrate in the models that maintaining global precipitation levels (3) requires less sulfate aerosol injection than maintaining temperature (2). The authors also point out that in the long run, without geoengineering, the temperature-dependent part of the precipitation response will dominate, and so the worst impacts of climate change we would be just yet to see. In the CDR scenario of net $co_2$ reduction (4) they find a wettening in both ESMs and an overall very similar precipitation response, however, this turns out to be a coincidence, because the temperature response is not so similar, which drives the precipitation response, and these are the fast response components to other forcing agents that seem to "compensate for" the difference. The ESMs also differ in other aspects. The temperature-dependence of the precipitation is more sensitive in the MPI-ESM - while the fast response (per unit forcing) are surprisingly similar - which is why stronger SRM is needed in MPI-ESM to maintain the temperature (2). The regional responses, both temperature, and precipitation, are very different between the models; it seems hopeless to predict in any location even the sign of the side-effect!!! Although it would not be completely useless to be able to predict at least bounds on the magnitude of the possible change.

The paper is written fairly decently. Although there are some repetitions, and the mathematical symbols and notation should be consistent. I attach an annotated version of the manuscript with corrections, comments, and questions. Overall, i found the paper a very worthwhile read. Even if i had my own experience with the drying under geoengineering and the fast components of the precipitation response, i have not been aware of the link of these two, and i did not know that the slow response can be put down approximately as a temperature-dependent component. The numerical work handling the EMSs was clearly a great effort too.

I have just a few perhaps/hopefully minor points to make.

1. Why did you base your four scenarios (1)-(4) on the RCP4.5 emission scenario when it remains within the Paris Agreement as represented in both ESMs, even within 1.5 K warming? I mean why not consider a business as usual scenario when we would much rather need geoengineering?

2. Regarding the control method for (2) and (3): You change the forcing level in the ESMs year-to-year, if needed, but is this forcing realistic, would it be consistent with changing the sulfate injection rate year to year? Is there not a transient effect? You wrote that the aerosol model ECHAM-HAMMOZ took 2 years spinup runs.

3. It seems to me that Fig. 7 does not verify that you have a temperature-dependent precip response component. But rather Fig. 4 does. In this respect, a more clear wording is needed at relevant points in the text.

Note: My policy is to not make a recommendation to editors on the publication of manuscripts. (Please consider my selection of the recommendation "minor revision" void, which i did only to be able to submit my review.) It is the editor's duty to make up their mind based on (ideally factual) referee reports, or one that reflects the referee's (ideally unbiased) opinion.

Please also note the supplement to this comment:
https://www.earth-syst-dynam-discuss.net/esd-2019-48/esd-2019-48-RC2-supplement.pdf

[Figure]

**Supplement:**

[revised manuscript text omitted]

---

## Author Comment (AC1) · 22 Jan 2020

We thank Anonymous Referee #1 for suggestions and comments which improved the manuscript. Our point by point answers to the comments are presented below. Referee comments are in bold and our replies in body text.

**General comments**
**The authors studied the climate responses to two different geoengineering approaches: sulfur injections and carbon dioxide removal. Using the Gregory regression approach, the authors separated temperature-independent (fast adjustment) and temperature-dependent (slow responses) components and then quantified their contributions to the total precipitation change in an RCP based scenario. The authors also compared the total amount of aerosols needed under various scenarios and the spatial pattern changes. The paper is well written and contributes to our understandings of the hydrological cycle responses to different geoengineering forcing. I would recommend publication after the authors addressed the following comments.**

**Specific comments**
**My understanding is that during the simulations using MPI-ESM and CESM, you directly used the "equilibrium" aerosol distributions patterns estimated from mECHAMHAMMOZ under the 1, 2, 3, 4, 5, and 6 Tg(S)/yr emission rates (in Fig.8 you used discrete injected sulfur rates). In the real world, if you change the emission rate of aerosols in year N, it takes time to adjust to a new equilibrium state. Therefore, there is a difference between your simulations and the real-world implementations (maybe it's worth to point out?). Do you have any idea how much the difference could be?**
This is true and was briefly mentioned in P7 L195, but we agree that this needs to be discussed more.

Thus in section 2.2 the original sentence :
"*An approximation inherent in this approach is that transitory ramp-up and ramp-down periods in the stratospheric aerosol burden with 1 Tg(S)/yr changes in SRM are not taken into account.*"'
is followed by:
"*Thus the simulated SRM changes take place faster than would occur in the real world. For example, the ECHAM-HAMMOZ simulation with 5 Tg(S)/yr injections requires 6 months to achieve 70% of the ultimate steady state aerosol optical depth (AOD) (533nm) after starting from background conditions.. When sulfur injections are suspended in the ECHAM-HAMMOZ simulation, the AOD decreases by roughly by 40% over the course of the first year. However, since the sulfur changes in our ESM simulations are only ±1 (Tg(S)/yr)) and do not usually occur in consecutive years we can assume that neglecting this time lag does not significantly alter our overall results.*"

**The discussion of regression in Section 3 is good. But in general, I think that the authors could add more information to support their statements and help the reader to better understand. For example, it would be very helpful if the author could show numbers at least for those regression slopes. It's hard to identify and compare these regression lines**

**with statements in the paper by eyes. Another example is shown in lines 258 to 260: the authors stated that "There was no large difference in surface albedo change between models" and "clear-sky SW absorption was linearly dependent on surface temperature". Can you remind me where I can find figures or data for such statements?**

We have now added the regression slopes to figures 2-4. In addition, we have now added similar regression figures for clear sky shortwave absorption and albedo changes to the supplemental material.

[Figure]

Figure 2: Gregory plots of the...Corresponding radiative forcing (intersection of linear fit and the y-axes (T=0)) in MPI-ESM and in CESM are shown in top and slope of the linear fit in bottom of legends next to the figure...

[Figure]

Figure 3: Gregory plots of the...Corresponding all-sky radiative forcing (intersection of linear fit and the y-axes (T=0)) in MPI-ESM and in CESM are shown in top and slope of the linear fit in bottom of legends next to the figure...

[Figure]

Figure 4: Gregory plots of the...and slope of the linear fit (shown in bottom of the legends) corresponds the slow response due to the temperature change...

[Figure]

Figure S1: Gregory plots of albedo change for a) MPI-ESM and b) CESM. Markers indicate a single-year global mean value for one ensemble member and solid lines are linear fits. Origin represents zero temperature and albedo anomaly compared to the Preind simulation.

[Figure]

Figure S2: Gregory plots of clear sky atmospheric absorption change for a) MPI-ESM and b) CESM. Markers indicate a single-year global mean value for one ensemble member and solid lines are linear fits. Origin represents zero temperature and atmospheric absorption anomaly compared to the Preind simulation.

**For each of the two models, the authors used regression results from three 20-year simulations (preindustrial, 2xCO2, and 4xCO2) and applied a logarithmical fit to estimate the coefficient "c" (the fast CO2 effect to precipitation) (lines 362 to 364). You have only three data points during the curve fit, so I am wondering how good the curve fit is? The fast precipitation response to CO2 change not only relates to the CO2 radiative effect but is also partly attributed to the physiological effect. It's fine to use the logarithmical form (I think the first paper used this relationship is in Cao et al., (2015)), but I think you might want to point it out here.**

This is a good comment and we agree that this should be pointed out. Using the logarithmical form requires the assumption that fast response depends linearly on absorbed radiation, but this was not shown or discussed. Based on e.g figure 2 in Samset et al 2016 this dependence is fairly linear, but some deviation occurs which is caused by, as the reviewer pointed out, physiological effects and also sensible heat flux. To address this comment we have now added a figure similar to that in Samset et al 2016, but using data from our simulations:.

[Figure]

**Figure S3.** Regression of fast precipitation response versus atmospheric absorption in a)MPI-ESM and b) CESM. R is the Pearson correlation coefficient.

Lines 352-363 have now been rewritten to discuss this further:

*"Based on our component analysis simulations we see that the fast precipitation response varies fairly linearly with absorbed radiation (See Fig. 3 in supplementary), but some deviation occurs due to changes to sensible heat flux and physiological responses of vegetation (DeAngelis et al., 2016). This result is consistent with that of Samset et al. (2016) and Myhre et al. (2017). The higher correlations in our simulations compared to Samset et al. (2016) may be due to the use of the fixed Sea Surface Temperature (SST) method to define the fast response in Samset et al. (2016): fast responses quantified with fixed-SST methods include land temperature adjustments.*

*Radiative forcings are generally assumed to be additive (Marvel et al., 2015). If we assume based on supplementary Fig. S3 that the overall fast response depends only on absorbed*

*radiation, it follows that the fast responses of individual forcing agents are also additive. In Sect. 3.3 we also showed that the slow temperature-dependent component does not depend on the applied forcing. We can thus describe the global mean precipitation change as the sum of the temperature-dependent slow component (a×ΔT) and all fast components (Fläschner et al., 2016):*

$$\Delta P = a \times \text{a}\Delta T + b(\text{SRM}) + c \times \ln \frac{\text{CO}_{2 \text{ preind}} + \Delta \text{CO}_2}{\text{CO}_{2 \text{ preind}}} + \text{BG},$$

*where a, and c are model-specific coefficients, b is a function of the SRM level, T is the simulated global mean surface temperature, $CO_{2 \text{ preind}}$ is the preindustrial CO2 concentration, $\Delta CO_2$ is the atmospheric $CO_2$ change relative to the preindustrial value, and BG is the background fast component, assumed to be the same for all scenarios. Coefficient a is obtained from the scenario-ensemble mean slope in Fig 4 (2.53 %/K for MPI-ESM and 2.27 % for CESM), while b is the fast component (intercept) from simulations of the corresponding SRM scenario (see ΔP(T=0) values in Fig 4). To calculate coefficient c, we again assume that fast precipitation response is linearly dependent on absorbed radiation. Radiative forcing due to $CO_2$ varies logarithmically with concentration (Etminan et al., 2016) and thus the fast precipitation response for $CO_2$ is also assumed to be logarithmically dependent on CO2 concentrations (see supplementary material Fig. S4)"*

We have also added a figure demonstrating this logarithmical fit to supplementary material (Fig 4).

[Figure]

Figure S4. Logarithmical fit of $CO_2$ fast precipitation calculated from preind, 2x$CO_2$ and 4x$CO_2$ scenarios.

**Technical corrections**
The following technical corrections have been made.
**Line 34: Add "increase" to "maintain the global mean temperature increase within 2∘C . . ."**
**Line 39: When the author says "a reduction of energy use", do you mean energy use associated with fossil fuel burning only? Because energy consumption provided by renewables and biofuels will not deteriorate the climate warming target.**
We rephrased the sentence to "*would require fast and significant reduction in use of fossil fuels complemented with carbon dioxide removal.*"
**Line 47: Missing ")" here: "(Luderer et al., 2018, IPCC 2018)"**
**Line 67: ". . . SRM would decrease global mean precipitation through the direct radiative effect described above". The hydrological cycle responses to solar geoengineering depend on magnitudes of geoengineering deployments and which case you are comparing with (the pre-industrial or high-CO2 world without SRM). I think the statement here should be more specific.**
This line is now rewritten:
"*One fundamental problem is that compensating GHG-induced warming with SRM would decrease global mean precipitation through the direct radiative effect described above.*"

**Line 80: add "for" to: ". . . continue to change for decades. . ."**
**Line 89: "separate" to "separates", "totally" to "total"**
**Line 94: "a" to "the"**
**Line 134: delete "the"**
**Line 182: what do you mean of "business as usual scenarios" here?**
"business as usual scenarios" is changed to "high emission scenario (RCP 8.5)"

**Line 196: add "into" to "taken into account"**
**Line 217: add "as"**
**Line 224: "were" to "was"**
**Line 232: Please define "Preind" before using it**
Rewritten as:
"*First, we calculated the clear-sky shortwave flux and temperature anomaly compared to the stable preindustrial conditions (Preind simulation) for each year individually, and performed a linear regression between the two variables*"
**Line 293: "warned" to "warm"**
**Line 294 and 325: the font is different**
**Line 315: "Fig.1" should be "Fig.6"**
**Line 324: "2010-2020" should be "2080-2100"**
**Line 355: there is an additional "a" in Eq. (1)**

---

## Author Comment (AC2) · 22 Jan 2020

We thank Tamas Bodai for thorough review, suggestions and comments. Our point by point answers to the comments are presented below. Referee comments are in bold and our replies in body text.

**The authors model two geoengineering methods, the Solar Radiation Management (SRM) and Carbon Dioxide Removal (CDR), in two Earth system models, the CESM and the MPI-ESM, and run simulations under several scenarios and analyse the global mean and regional temperature and precipitation responses. The considered scenarios are the following: (1) RCP4.5 (2) RCP4.5 with SRM controlled to approximately maintain present-day global mean temperature throughout the 21st c. (3) RCP4.5 with SRM controlled to approximately maintain present-day global mean precipitation throughout the 21st c. (4) RCP4.5 with an ambitious CDR removing 1% of CO2 concentration per year They give some evidence in both ESMs that the global mean precipitation response has a global mean temperature-dependent component, which is based on the methodology proposed by Gregory et al. (2004). Besides this the precipitation can respond very fast, at least to the greenhouse co2 forcing, and this fast response is of opposite sign compared with the temperature-dependent part. To aerosol forcing the fast response is negligible. This is the reason why there is less precipitation in an SRM scenario when global warming is fully compensated. Conversely, the authors make the point and demonstrate in the models that maintaining global precipitation levels (3) requires less sulfate aerosol injection than maintaining temperature (2). The authors also point out that in the long run, without geoengineering, the temperature-dependent part of the precipitation response will dominate, and so the worst impacts of climate change we would be just yet to see. In the CDR scenario of net co2 reduction (4) they find a wettening in both ESMs and an overall very similar precipitation response, however, this turns out to be a coincidence, because the temperature response is not so similar, which drives the precipitation response, and these are the fast response components to other forcing agents that seem to "compensate for" the difference. The ESMs also differ in other aspects. The temperature-dependence of the precipitation is more sensitive in the MPI-ESM - while the fast response (per unit forcing) are surprisingly similar - which is why stronger SRM is needed in MPI-ESM to maintain the temperature (2). The regional responses, both temperature, and precipitation, are very different between the models; it seems hopeless to predict in any location even the sign of the side-effect!!! Although it would not be completely useless to be able to predict at least bounds on the magnitude of the possible change.**

**The paper is written fairly decently. Although there are some repetitions, and the mathematical symbols and notation should be consistent. I attach an annotated version of the manuscript with corrections, comments, and questions.**

These are now corrected based on the referee's comments in the annotated version.

**Overall, i found the paper a very worthwhile read. Even if i had my own experience with the drying under geoengineering and the fast components of the precipitation response, i have not been aware of the link of these two, and i did not know that the slow response can be put down approximately as a temperature-dependent component. The numerical work handling the EMSs was clearly a great effort too. I have just a few perhaps/hopefully minor points to make.**

**1. Why did you base your four scenarios (1)-(4) on the RCP4.5 emission scenario when it remains within the Paris Agreement as represented in both ESMs, even within 1.5 K warming? I mean why not consider a business as usual scenario when we would much rather need geoengineering?**

Actually RCP4.5 does not remain within the Paris Agreement targets (less than 1.5 or 2 K warming compared to the preindustrial era). Figure 6 was misleading in this respect because the plotted temperature anomaly is calculated with respect to 2010-2020 and not with respect to the preindustrial era. We have now included text ("Temperature/precipitation anomalies relative to 2010-2020") to the y-axis labels to make this point more clear for readers. In addition we now rewrite Line 316 as: "*Under RCP45, the global mean temperature increased by 1.30 K and 1.20 K over the 2010-2020 average in MPI-ESM and CESM, respectively.*"

Likewise, L329-331 now reads: "*Thus, in both ESMs the SRM-PRECI global temperature increase (~1.64 K in MPI-ESM and ~1.27 K in CESM compared to the preindustrial average) stayed within the 2 C target of the Paris Agreement. For CESM, the SRM-PRECI temperature increase also stayed within the 1.5 C Paris target.*"

[Figure]

Figure 6: Global mean temperature anomalies in a) MPI-ESM and b) CESM, and global mean precipitation anomalies in c) MPI-ESM and d) CESM. Numbers to the right of each figure indicate the global mean difference between 2080-2100 and 2010-2020. Shaded areas show the maximum and minimum across three ensemble members.

**2. Regarding the control method for (2) and (3): You change the forcing level in the ESMs year-to-year, if needed, but is this forcing realistic, would it be consistent with changing the sulfate injection rate year to year? Is there not a transient effect? You wrote that the aerosol model ECHAM-HAMMOZ took 2 years spinup runs.**

There is indeed a transient effect which is not taken into account in the simulations. This was mentioned in section 2.2, but is now discussed in more detail as follows:

*"An approximation inherent in this approach is that transitory ramp-up and ramp-down periods in the stratospheric aerosol burden with 1 Tg(S)/yr changes in SRM are not taken into account. Thus the simulated SRM changes take place faster than would occur in the real world. For example, the ECHAM-HAMMOZ simulation with 5 Tg(S)/yr injections requires 6 months to achieve 70% of the ultimate steady state aerosol optical depth (AOD) (533nm) after starting from background conditions.. When sulfur injections are suspended in the ECHAM-HAMMOZ simulation, the AOD decreases by roughly by 40% over the course of the first year. However, since the sulfur changes in our ESM simulations are only ±1 (Tg(S)/yr)) and do not usually occur in consecutive years we can assume that neglecting this time lag does not significantly alter our overall results."*

**3. It seems to me that Fig. 7 does not verify that you have a temperature-dependent precip response component. But rather Fig. 4 does. In this respect, a more clear wording is needed at relevant points in the text.**

Based on the referee's comments in the supplement, it appears that we were not sufficiently clear in stating that the temperature-dependent component depends only on temperature and not on the forcing driving the temperature change. This was mentioned on L57 in the introduction and at L309-L310 but is also now articulated more clearly in section 3.3. We have also clarified the rationale for our use of the component-based approach (i.e., in Fig 7). A new figure illustrating the dependence of the fast precipitation response on absorbed radiation has also been included in the supplement. Sect 4.2 it now reads:

*"Based on our component analysis simulations we see that the fast precipitation response varies fairly linearly with absorbed radiation (See Fig. 3 in supplementary), but some deviation occurs due to changes to sensible heat flux and physiological responses of vegetation (DeAngelis et al., 2016). This result is consistent with that of Samset et al. (2016) and Myhre et al. (2017). The higher correlations in our simulations compared to Samset et al. (2016) may be due to the use of the fixed Sea Surface Temperature (SST) method to define the fast response in Samset et al. (2016): fast responses quantified with fixed-SST methods include land temperature adjustments.*

*Radiative forcings are generally assumed to be additive (Marvel et al., 2015). If we assume based on supplementary Fig. S3 that the overall fast response depends only on absorbed radiation, it follows that the fast responses of individual forcing agents are also additive. In Sect. 3.3 we also showed that the slow temperature-dependent component does not depend on the applied forcing. We can thus describe the global mean precipitation change as the sum of the temperature-dependent slow component (a×ΔT) and all fast components (Fläschner et al., 2016):*

[Figure]

**Figure S3.** Regression of fast precipitation response versus atmospheric absorption in a) MPI-ESM and b) CESM. R is the Pearson correlation coefficient.

Along similar lines, we feel that inclusion of the BG component in Fig 7 could be misleading in giving the impression that the agreement between total modelled precipitation and the sum of the individual components arises from the BG component. While this is obviously true in the case of RCP 4.5 (from which the BG component is calculated as the residual between equation 2 (previously 1) and modelled precipitation), the same BG component (calculated based on RCP 4.5) is then used as a component in all geoengineering scenarios. Thus in those cases the precipitation agreement referred to above does not arise from the BG component.

To prevent any such confusion, we added a supplemental figure analogous to Fig. 7 but with yearly precipitation differences in the geoengineering scenarios compared to the corresponding year in RCP4.5. Doing the comparison in this way removes the BG component. As this figure shows, the component-based precipitation agrees well with the total modelled quantity in this framework as well.

The corresponding section has been rewritten to make this more clear for the reader:
*"Figure 7 shows the precipitation component for each scenario in MPI-ESM and CESM. In general, the precipitation signal as estimated by the fast and slow components via Eq. (2) corresponds well to the actual model quantity in both ESMs for all scenarios. For RCP4.5 this is*

*an obvious result because the BG is derived as the residual between the modelled precipitation and the sum of the individual components for this very scenario. However, the component-based and full precipitation signals also agree well for the other scenarios, even though the BG component is calculated from the RCP45 case. From year-2020 to year-2100 the mean differences between the Eq. (2) results and the actual model quantities were ranged from -0.01% to 0.04% for MPI-ESM and from -0.16% to 0.05% for CESM. Fig. S5 in supplementary material shows the precipitation responses under the geoengineering scenarios as anomalies relative to the RCP4.5 case. The plotted precipitation differences in Fig. S5 are thus independent of the BG component. We see from this figure that the individual components can be reliably used to understand the drivers of precipitation change for each scenario."*

We choose to include the BG component in figure 7 for the following reasons
1) For clarify we wish the modelled precipitation under the individual scenarios to be the same in both figures 6 and 7.
2) The BG component shows that there are several other significant fast precipitation components (in the background) which are affecting precipitation in RCP 4.5 scenario
3) The BG component was clearly different in same scenario modelled by MPI-ESM and CESM. This is one of the main reasons for the diversity in climate models precipitation results found here.
4) We wanted to show and highlight that component-based analyses are not significantly dependent on the background. For example, the precipitation difference between RCP4.5 and CDR can be represented by the fast $CO_2$ component and the temperature-based component for years 2020 and 2100, even though the background atmospheric conditions have changed.